Relationship between lower extremity strength asymmetry and linear multidimensional running in female tennis players

Turkeri Cenab 1
Oztürk Bariscan bariscan.ozturk.bc@gmail.com 1
Koç Murat muratkoc@erciyes.edu.tr 2
Engin Hakan 1
Uluöz Eren 1
Yılmaz Cem Yoksuler 1
Özsu Banu Nurdan 1
Celik Lutfi Tolga 1
Şeker Mehmet Emin 1
Çiçek İsmail 1
Uzunca Caner 1
Bahçivan İbrahim 1
Abbass Ahmed Abdelmoeen 3
1 Faculty of Sport Sciences, Çukurova University , Adana , Turkey
2 Faculty of Sport Sciences, Erciyes University , Kayseri , Turkey
3 Faculty of Physical Education, Benha University , Benha , Egypt
Espada Mário
Electronic publication date: 2024 Sep 26
Publication date: 2024
Volume: 12
Electronic Location ID: e18148
Received 2024 May 16; Accepted 2024 Aug 30
Copyright: ©2024 Turkeri et al.
Copyright year: 2024
Copyright holder: Turkeri et al.
License: This is an open access article distributed under the terms of the Creative Commons Attribution License, which permits unrestricted use, distribution, reproduction and adaptation in any medium and for any purpose provided that it is properly attributed. For attribution, the original author(s), title, publication source (PeerJ) and either DOI or URL of the article must be cited.
License URL: https://creativecommons.org/licenses/by/4.0/

Keywords: Inter Limb Asymmetry, Tennis, Linear and Multidimensional Running

Funding: The authors received no funding for this work.

==============================
Background

Tennis requires movement abilities in changing playing situations. This article investigates the relationship between lower extremity strength asymmetry ratio and linear and multidimensional running performances in female tennis players.

Methods

A total of 56 female tennis players, with a mean age of 15.44 ± 0.50 years, participated in the study—the research design involved three sessions at 48-hour intervals. In the first session, athletes performed dominant and non-dominant countermovement jump (CMJ) and board jump (BJ) tests. The second (sec) session included 10-meter (−m) and 30-m linear running performance tests, while the final session assessed multidimensional running performance with a change of direction (COD) test. The relationship between CMJ and BJ asymmetry ratios and linear and multidimensional running performances was analysed using the Pearson correlation coefficient. Bilateral asymmetry rates in linear and multidimensional running performance were determined through linear regression analysis.

Results

The dominant CMJ recorded 17.56 ± 3.47 cm, while BJ was 130.23 ± 21.76 cm, and the non-dominant CMJ measured 16.79 ± 4.51 cm with a BJ of 147.52 ± 30.97 cm. The athletes had a CMJ asymmetry rate of 12.67 ± 11.29% and a BJ asymmetry rate of 7.19 ± 5.28%. A relationship was seen between the CMJ asymmetry rate and 30-m running performance (r = 0.368, p < 0.05). There was no correlation between BJ asymmetry rate and 10-m running performance. Significant correlations were found between 30-m (r = 0.364) and COD (r = 0.529) running performances (p < 0.05).

Conclusions

It can be said that the CMJ asymmetry ratio may negatively affect 30-m and the BJ asymmetry ratio may negatively affect 30-m and COD performance.

Introduction

Muscle strength and power asymmetry refer to variations in strength and power between limbs during a specific joint action. Typically, these variances are assessed by comparing the following: strong versus weak limbs, dominant versus non-dominant limbs, or injured versus non-injured limbs (Bond et al., 2017). Humans lean to use one side of the body more than the other for motor performance. This learning is due to one hemic being dominant over the other, called cerebral dominance (CD hereafter). Vosburg et al. (2022) argued that coordinating the strength ratios is more important because the glitch in opposing muscles may increase the risk of instability and musculoskeletal injury. In recent years, more academics have argued that extremity asymmetry has more impact on injury prevalence and performance (Bishop, Turner & Read, 2018; Helme et al., 2023; Maloney, 2019).

Recent literature suggests that limb asymmetries, particularly those involving the lower extremities, may elevate the risk of injury (Vosburg et al., 2022; Yalfani & Raeisi, 2022). However, these conclusions are predominantly based on indirect methods, and there is a significant lack of studies that directly investigate the impact of asymmetry on specific performance outcomes (Afonso et al., 2022). Dynamic sports, such as tennis, which necessitate frequent movement and direction changes, offer a critical context for examining the effects of limb asymmetries on performance. Tennis requires various performance metrics, including speed, change of direction (COD), and explosive movements. Despite this, existing research has largely concentrated on general sports performance and injury risk, with limited studies exploring how asymmetry specifically affects tennis-related performance (Chapelle et al., 2022; Espada et al., 2023; Fort-Vanmeerhaeghe et al., 2022; Michailidis et al., 2020; Nicholson et al., 2022; Raya-González, Clemente & Castillo, 2021; Villanueva-Guerrero et al., 2024).

Additionally, it has been suggested that limb asymmetries may elevate the risk of injury in female athletes, as women frequently display greater kinematic asymmetries compared to men (Hewett et al., 2005). Sports such as tennis may exacerbate these asymmetries due to uneven mechanical loading on the lower extremities (Chapelle et al., 2022). However, current studies tend to focus on the general risk of injury rather than the specific impact of asymmetry on performance, particularly in tennis. Therefore, it is essential to conduct research that directly investigates the effects of limb asymmetry on performance in tennis players. This approach would enhance our understanding of how asymmetry influences specific performance metrics, such as linear speed and change of direction (COD), rather than overall tennis performance.

This study is noteworthy as one of the pioneering investigations into the impact of limb asymmetry on performance, specifically among female tennis players. It addresses existing gaps in the literature by examining the effects of lower extremity strength asymmetry on linear speed and change of direction (COD). In doing so, it provides valuable insights into how limb asymmetry affects these specific performance metrics and offers gender-specific information that could inform strategies to enhance both athletic performance and coaching practices.

Materials & Methods

Research design

A correlational research design was employed to examine the relationship between tennis players’ lower limb asymmetry rates and their linear and directional running performances. This design is used to explore the relationships between two or more variables and to provide insights into potential cause–effect processes (Büyüköztürk et al., 2008).

Human ethics

This study was conducted by the tenets of the Declaration of Helsinki and approved by the Ethics Committee of the Benha University Clinical Research (approval number: 12.7.2022; approval date: 04 September 2022 (MoHP No: 0018122017/Certificate No:1017)). Informed consent forms were verbally communicated, and each participant was asked to sign a copy.

Sample size

The sample size of the study was calculated as α = 0.05, Power = 0.95 (1-ß) in the G*Power program (ver 3.1.9.2) and the number of participants for linear regression analysis from the Exatc analysis group was n = 49. To prevent data loss and increase the validity of the research, the number of participants was determined as the total number of tennis players in the region (Benha City) (n = 69). Athletes who did not meet the inclusion criteria (n = 13) were excluded from the study. The research was completed with 56 female athletes.

Participants

A total of 56 female tennis players participated in the research. The sports age of the athletes is 4.48 ± 1.53 years, age is 15.44 ± 0.50 years, height is 1.66 ± 0.04-m (meter), body mass is 63.87 ± 4.85 kg and BMI is 19.89  ± 1.71 kg/m2. Before the research, the athletes’ injuries were questioned. Athletes with injuries were not included in the research. Since it is thought that the menstrual period (between 1 and 28 days) may affect anaerobic performance, athletes during the menstrual period (between 1 and 5 days) were not included in the study (Cook, Kilduff & Crewther, 2018). The study inclusion and exclusion criteria are shown in Fig. 1. All tests of the athletes included in this study were performed under the supervision of an expert from the university’s movement and training sciences department.

Figure 1 Research flow chart.

Procedure

Field-based tests offer several advantages, as they appear to better mimic the specific demands of complex sports such as tennis (Ferrauti et al., 2013). Additionally, field-based tests are easy to implement, ecologically valid, and time-efficient because they allow for the simultaneous recording of multiple players and typically require inexpensive equipment (Ulbricht, Fernandez-Fernandez & Ferrauti, 2013). As a result, field-based tests such as 20-m linear sprints, COD tests, and jumping ability tests (e.g., countermovement jump [CMJ]) are commonly used among tennis players. A distance of 20-m is recommended for testing tennis players (Kovacs, 2006) although research has shown that sprint performance (e.g., 20-m time) is largely dependent on maximum horizontal power output during sprint acceleration. Some players may not reach maximum speed within the given distance, potentially increasing running speed theoretically (Volk, Vuong & Ferrauti, 2023). Consequently, the extrapolated linear force-velocity (F-v) and parabolic power-velocity (P-v) relationships can accurately describe the overall mechanical capacity to produce horizontal force and velocity at both low and high speeds during sprint running (Samozino et al., 2016). Therefore, the 30-m linear sprint test was used to allow participants to reach maximal speed. However, there is a paucity of research directly investigating the relationship between the specific tests used in this study and both tennis performance and injury risk. Future studies are recommended to examine these relationships in more depth.

The study consisted of three sessions at 48-hour intervals. In the first session of the study, single-leg dominant and non-dominant CMJ and BJ tests were applied to the athletes. In the second session, 10-m and 30-m sprint tests were applied to determine the linear running performance of the athletes. In the last session of the study, the COD test was applied to determine the multidimensional running performances. Before all sessions, the athletes were given 5-min (minutes) of jogging, 10-min of stretching and a general warm-up under the direction of their coach. In addition, the study was carried out by the same researcher at the same time of the day (17:00–18:00) to eliminate the effect of the circadian rhythm on the athletes (Öztürk et al., 2022; Öztürk et al., 2023). The flow chart of the research is shown in Fig. 1, and the design of the research is shown in Fig. 2.

Figure 2 Research design.

Data collection

Body mass and height measurement

The athletes’ height (with 0.5 cm sensitivity) and body mass (with 100 g sensitivity) were measured using a Seca stadiometer. The demographic characteristics of the participants (age and sports age) were determined by the questionnaire form prepared by the researchers.

Inter limb asymmetry tests

Single leg counter movement jump

Athletes’ dominant and non-dominant leg (unilateral) CMJ tests were measured using a Witty Microgate jumping mat. Athletes jumped to the highest point at the midpoint of the jump mat, where they could jump with their hands-free. During the measurement, the jump height of the athlete was recorded. The measurement was repeated three times, and the average time was recorded.

The time of flight method in the CMJ test is regarded as a reliable measurement technique when performed with free-arm movements (Glatthorn et al., 2011). However, there are certain limitations concerning the validity of this method. Specifically, the free use of arms may introduce kinematic variability, potentially impacting the validity of the test results (Markovic & Jaric, 2007). Therefore, while the findings of this study are considered reliable, further research is warranted to better establish the validity of the test.

Single leg broad jump

Athletes’ dominant and non-dominant leg broad jump tests (BJ) were measured on a pre-prepared jumping track. Athletes performed dominant and non-dominant BJ from the determined starting point to the farthest point they could jump. After the jump, the closest distance from where they fell to the starting point was measured. Three jump tests were performed for each athlete, and the average result was recorded (Impellizzeri et al., 2007).

Asymmetry analysis

The values obtained from the athletes’ vertical and horizontal jump performances with their dominant and non-dominant legs were used to determine their asymmetry (Impellizzeri et al., 2007). The asymmetry formula is calculated as follows. Dominant−Non DominantDominant×100.

Linear sprint test

10–30 m linear sprint test

The 10–30 m sprint test was measured using the Witty Microgate photocell system. The test was carried out on a 30 m track. Measurements were taken with photocells placed 10-m and 30-m away from the starting line. Athletes take their place at the starting point, leaving 1-m behind the starting photocell. Athletes were asked to run a distance of 30-m at maximum speed, and 10-m and 30-m values were recorded. The measurement was repeated 3 times and the average time was recorded (Meylan & Malatesta, 2009).

Multi-dimensional running test-change of direction test

The COD test was measured using the Witty Microgate photocell system. The test consists of three slaloms placed at a distance of 5-m as Zig-Zag at an angle of 100° to each other at a distance of 20-m. The athletes passed the test between three slaloms at the highest speed, starting 1-m behind the starting line. The test was applied three times for each athlete, and the best score was recorded (Loturco et al., 2016; Pereira et al., 2018).

Statistical analysis

SPSS 22.0 package program was used for statistical analysis. Skewness and Kurtosis test results were examined for normal distribution of the data. In this analysis, the normality assumption was accepted between −1.5 and +1.5, according to Tabachnick and Fidell (Tabachnick, Fidell & Ullman, 2013). The presence of outliers was examined using standardized residuals and box plots. Since no outliers were found that would significantly distort the results, none were removed from the analyses. According to the normality test, it was determined that the data showed normal distribution. The Pearson correlation coefficient determined the relationship between lower limb asymmetry and linear and multidimensional running performances. Additionally, the effect of bilateral asymmetry (horizontal and vertical) ratio on linear and multidimensional running performance was determined by regression analysis. In this study, the significance level was accepted as p < 0.05.

Results

The mean age of the tennis players was 15.44 ± 0.50 years, with a height of 1.66 ± 0.04 m, body mass of 63.87 ± 4.85 kilograms, and a training age of 4.48 ± 1.53 years. Additionally, the BMI of the tennis players was 19.89 ± 1.71 kg/m2 (Table 1).

Table 1 Demographic characteristics of female tennis players.

	n	Min.	Max.	x¯	Std. Deviation	
Age (year)	56	15,00	16,00	15,44	0,50	
Height (m)	56	1,60	1,75	1,66	0,04	
Body Mass (kg)	56	52,00	73,70	63,87	4,85	
BMI (kg/m2)	56	15,19	24,69	19,89	1,71	
Training Age (year)	56	2,00	8,00	4,48	1,53	
Notes.

BMI Body mass index

Linear running performances were measured as 2.14 ± 0.19 s for 10-m and 5.02 ± 0.58 s for 30-m. Multi-dimensional running performance, measured as change of direction (COD), was 6.94 ± 0.42 s. (Table 2).

Table 2 Linear and multi dimensional running results of female tennis players.

	n	Min.	Max.	x¯	Std. Deviation	ICC(%95)	
10 m (sec)	56	1,75	2,56	2,14	0,19	0,987	
30 m (sec)	56	4,19	6,47	5, 02	0,58	0,958	
COD (sec)	56	6,21	7,62	6,94	0,42	0,943	
Notes.

COD Change of direction test

For the dominant leg, the CMJ was 17.56 ± 3.47 cm and the BJ was 130.23 ± 21.76 cm. For the non-dominant leg, CMJ was 16.79 ± 4.51 cm and BJ was 147.52 ± 30.97 cm. The bilateral asymmetry ratios were 12.67 ± 11.29 for CMJ asymmetry and 7.19 ± 5.28 for BJ asymmetry (Table 3).

Table 3 Dominant, non-dominant, CMJ and BJ results and bilateral asymmetry ratio of female tennis players.

		n	Min.	Mak.	x¯	Std. Deviation	ICC(%95)	
Dominant	CMJ (cm)	56	9,61	23,74	17,56	3,47	0,985	
BJ (cm)	56	71,00	180,00	130,23	21,76	0,961	
Non Dominant	CMJ (cm)	56	7,06	26,50	16,79	4,51	0,968	
BJ (cm)	56	77,00	195,00	147,52	30,97	0,912	
Bilateral Asymmetry (%)	CMJ (cm)	56	0,00	39,51	12,67	11,29	0,986	
BJ (cm)	56	0,40	21,00	7,19	5,28	0,918	
Notes.

CMJ Countermovement jump

BJ Broad jump

A significant positive relationship was found between CMJ asymmetry ratios and 30-m running performance (r = 0.368; p = 0.005). However, no significant relationships were observed between CMJ asymmetry and 10-m running performance (r = 0.136; p = 0.316) or COD performance (r = 0.121; p = 0.375). Additionally, no correlation was found between BJ asymmetry ratios and 10-m running performance (r = −0.029; p = 0.834). Significant positive correlations were found between BJ asymmetry ratios and both 30-m (r = 0.364; p = 0.006) and COD (r = 0.529; p = 0.001) running performances (Table 4) (Fig. 3).

Table 4 The relationship between bilateral asymmetry rates and linear and multi dimensional running performances of female tennis players.

			10 m (sec)	30 m (sec)	COD (sec)	
Bilateral Asymmetry (%)	CMJ (cm)	r	0.136	0.368**	0.121	
BJ (cm)	r	−0.029	0.364**	0.529***	
Notes.

CMJ Countermovement jump

BJ Broad jump

* p < .05

** p < .01

*** p < .001

Figure 3 Correlation between vertical and horizontal jump asymmetry with 10-m, 30-m, and COD performance.

The analysis revealed that vertical asymmetry ratios significantly positively affected 30-m running performance, with each unit increase in vertical asymmetry ratio leading to a 0.019-second increase in 30-m running time, accounting for 13.5% of the variance in performance (Table 5). Similarly, horizontal asymmetry ratios significantly positively affected both 30-m and COD performances, with each unit increase in horizontal asymmetry ratio resulting in a 0.040-second increase in 30-m running time and a 0.043-second increase in COD performance, explaining 13.3% and 28.0% of the variance in these performances, respectively (Table 6).

Table 5 Effect of vertical asymmetry ratio on 10-m, 30-m and COD performance in female tennis players.

	β	Standart Eror	Beta	t	p	r	r 2	
10 m (sec)	0,002	0,002	0,136	1,012	0,316	0,136	0,019	
30 m (sec)	0,019	0,007	0,368	2,909	0,005**	0,368	0,135	
COD (sec)	0,005	0,005	0,121	0,895	0,375	0,121	0,015	
Notes.

COD Change of direction test

** p < 0.01

Table 6 Effect of horizontal asymmetry ratio on 10-m, 30-m and COD performance in female tennis players.

	β	Standart Eror	Beta	t	p	r	r 2	
10 m (sec)	−0,001	0,005	−0,029	−0,211	0,834	0,029	0,001	
30 m (sec)	0,040	0,014	0,364	2,872	0,006**	0,364	0,133	
COD (sec)	0,043	0,009	0,529	4,580	<0.001***	0,529	0,280	
Notes.

COD Change of direction test

** p < 0.01

*** p < 0.001

Discussion

This study investigates the relationship between lower extremity strength asymmetry and linear multidirectional running in tennis. These results indicate a positive relationship between the horizontal asymmetry ratio in COD and the 30-m linear sprint. Tennis is typically characterised by short bursts of activity lasting 5–10 s, followed by periods of recovery lasting 10–20 s. Most of the movements, such as jumping, sprinting, and changing direction, performed by tennis athletes are of high intensity and involve unilateral movement patterns (Fernandez-Fernandez, Sanz-Rivas & Mendez-Villanueva, 2009). Due to the prevalence of these unilateral movements, tennis players often exhibit an increased rate of bilateral asymmetry. This heightened asymmetry can elevate the risk of athlete injuries (Fort-Vanmeerhaeghe et al., 2022; Heil, Loffing & Busch, 2020; Raya-González, Clemente & Castillo, 2021). In addition, repetitive unilateral movements in training and competition conditions cause a decrease in the performance of athletes (Bishop et al., 2019; Raya-González, Clemente & Castillo, 2021). This situation has paved the way for sports scientists to focus on this area and research bilateral asymmetry, athletes’ performance, and injury risks, such as the idea (Kaffel et al., 2013) about suggesting strength and conditioning training programs focusing on preventive objectives. Recently, Bettariga et al. (2022) conducted a study showing that using unilateral strength programs reduces inter-limb asymmetry. When the studies are examined, it is seen that the studies generally compare the vertical jump asymmetry rates and change of direction performances of the athletes (Torreblanca-Martínez, Torreblanca-Martínez & Salazar-Martínez, 2020; Yeung et al., 2021). However, the literature reveals a lack of studies investigating strength asymmetries about linear and multidimensional running performances among tennis players. In this context, our study was carried out to determine the relationship between the strength asymmetry ratios of the athletes and their linear and multidimensional running performances, and to contribute to the lack of literature.

Bilateral asymmetry rates of tennis players

When the bilateral asymmetry rates of the tennis players participating in the study were analyzed as a percentage, it was found that the CMJ asymmetry rate was 12.67 ± 11.29 (%) and the Broad Jump asymmetry rate was 7.19 ± 5.28 (%). Bishop et al. (2019) found the vertical jump asymmetry rates of the athletes as 12.54% and the horizontal jump asymmetry rates as 6.79% in a study. Lockie et al. (2014) found the vertical asymmetry rates of the athletes as 10.4% and the horizontal asymmetry rates as 5.1%. In another study, Michailidis et al. (2020) found the vertical asymmetry rate as 10.38% and the horizontal asymmetry rate as 5.24%. The results in the literature are similar to the findings in our study. Tennis, generally lasting 5–10 s, and repetitive high-intensity activities with 10–20 s of recovery are mostly exhibited with a unilateral movement pattern (Fernandez-Fernandez, Sanz-Rivas & Mendez-Villanueva, 2009). In particular, the sprint, acceleration, deceleration, and jump performances applied in the game require the active activity of the lower extremities. During competitions or rallies, the force exerted by the athlete on the ground with one foot relative to the other during each stroke, known as the “kinetic chain”, varies between linear and multidimensional movements performed by the player (Roetert et al., 2009). This situation facilitates the development of greater strength in one limb compared to the other, leading to bilateral strength asymmetry in the athlete. Consequently, this imbalance may contribute to an increased risk of injury.

Relationship between CMJ and linear and multidimensional running performances

When the relationship between CMJ asymmetry rates and the linear and multidimensional running performances of the tennis players participating in the study was examined, a positive relationship was found between 30-m running performances (r = 0.368, p < 0.05). However, no relationship was found between 10-m (r = 0.136) and COD (r = 0.121) running performances (p > 0.05). Many studies have reported no relationship between the asymmetry rate obtained from the vertical jump test in trained athletes and their linear and multidimensional running performances (Chiang, 2014; Dos’Santos et al., 2017; Exell et al., 2017; Haugen et al., 2018; Lockie et al., 2017; Merino-Muñoz et al., 2021). In contrast to these studies, others have found different results from our findings (Bishop et al., 2021; Lockie et al., 2012; Ranisavljev, Matić & Janković, 2020; Sannicandro et al., 2011). However, the differences in the results can be attributed to variations in the individuals who participated in these studies and the use of different tests compared to our research.

While the vertical jump asymmetry rate did not appear to affect the 10-m sprint and COD, it did have an impact on the 30-m sprint. It can be speculated that asymmetry may have a greater effect on running performance as the distance increases. However, no research has been conducted on this, so it remains an assumption.

Relationship between BJ and linear and multidimensional running performances

When the relationship between BJ asymmetry rates and linear and multidimensional running performances of tennis players was examined, no relationship was found with 10-m (r = −0.029) running performance (p > 0.05). A significant positive correlation was found between 30-m (r = 0.364) and COD (r = 0.529) running performances (p < 0.05). Bishop et al. found that the results of the asymmetry in the jumping performances of the athletes negatively affect the multidimensional running performances (Bishop et al., 2022). Michailidis et al. (2020) found a positive correlation between horizontal jump (r = 0.26) asymmetry rate and multidimensional running performances in a study conducted with athletes. Accordingly, he stated that the increase in jump asymmetry rates negatively affects multidimensional running performance in athletes. Maloney (2019) in a study examining the relationship between jump asymmetry score and COD performance, found a positive relationship between the athlete’s asymmetry score and multidimensional running performance (r = 0.60) (p < 0.05). The results we found in our research show similarities with the studies in the literature. Tennis, by its nature, includes linear and multidimensional runs. In activities such as jumping and hitting, a higher level of load is placed on the lower extremity, which is usually dominant, simultaneously with the upper extremity. While the dominant extremity develops with repeated loading, the non-dominant extremity does not develop that much. This increases the difference in force that can be produced between both extremities. Bilateral strength differences can also put the athlete’s health at risk and cause injury because studies have shown that the increase in bilateral asymmetry causes injuries in athletes (Maloney, 2019). As a result of the findings, it can be said that the horizontal asymmetry rates of the athletes do not affect the 10-m linear running values but negatively affect the 30-m linear running and multidimensional running values.

It can be posited that since the 10-m linear running test is shorter in duration compared to the 30-m running test, it is less influenced by BJ asymmetry. Conversely, as the running distance increases, the impact of BJ asymmetry becomes more pronounced, leading to a decline in 30-m performance.

Practical application

Bilateral asymmetry not only affects athletic performance outcomes (such as 10-m, 30-m, and COD times) but can also lead to sports injuries. Therefore, in future tennis training, coaches and athletic performance trainers should closely monitor the asymmetry in both limbs of athletes’ bodies and the severity of this asymmetry. For athletes with significant limb asymmetry, coaches should use targeted training interventions to address and reduce these imbalances. Such measures can enhance overall performance and contribute to the prevention of sports-related injuries. Coaches and trainers should adopt advanced training methods to reduce asymmetry and promote balanced athletic development. These approaches should focus on increasing the efficiency and effectiveness of training programs.

Limitation

Although tennis is played as an upper extremity-dominant game, the lower extremity plays a crucial role in the mechanical elements of the game. By its nature, tennis is a high-intensity sport with sudden decelerations and accelerations. Therefore, acceleration and deceleration performance are important. However, only maximal sprint and direction change performances were included in this study. Acceleration and deceleration performance were not included in the research, which is a limitation of the performance outcomes for this study. Future research should consist of both linear and direction-changing acceleration-deceleration running tests. Another limitation of this research is that it only studied female athletes, particularly those entering puberty. Future studies should also include male athletes and examine potential differences between genders. Additionally, future research should consider including peak height velocity in participant selection, expanding the sample group for better generalisability of results, and considering age and performance level categories.

Conclusions

It was found that the CMJ asymmetry ratio did not affect 10-m and COD performance. However, it was determined that the CMJ asymmetry ratio could negatively affect 30-m running performance. It was also determined that BJ asymmetry may negatively affect athletes’ linear and multidimensional running performance. It is stated in the literature that unilateral strength training will have a positive effect. Considering all these factors, it can be recommended that practitioners apply tests to determine bilateral asymmetry scores and implement more unilateral strength training in favour of the weaker side to reduce or eliminate the determined asymmetry rate.

Supplemental Information

Supplemental Information 1 List of performance tests of athletes

The authors would like to thank the individuals who participated in this investigation.

Additional Information and Declarations

Competing Interests

Author Contributions

Human Ethics

Data Availability

The authors declare there are no competing interests.

Cenab Turkeri analyzed the data, authored or reviewed drafts of the article, and approved the final draft.

Bariscan Oztürk conceived and designed the experiments, performed the experiments, authored or reviewed drafts of the article, and approved the final draft.

Murat Koç conceived and designed the experiments, analyzed the data, authored or reviewed drafts of the article, and approved the final draft.

Hakan Engin performed the experiments, authored or reviewed drafts of the article, and approved the final draft.

Eren Uluöz performed the experiments, analyzed the data, authored or reviewed drafts of the article, and approved the final draft.

Cem Yoksuler Yılmaz conceived and designed the experiments, analyzed the data, authored or reviewed drafts of the article, and approved the final draft.

Banu Nurdan Özsu analyzed the data, prepared figures and/or tables, authored or reviewed drafts of the article, and approved the final draft.

Lutfi Tolga Celik analyzed the data, authored or reviewed drafts of the article, and approved the final draft.

Mehmet Emin Şeker conceived and designed the experiments, prepared figures and/or tables, authored or reviewed drafts of the article, and approved the final draft.

İsmail Çiçek performed the experiments, authored or reviewed drafts of the article, and approved the final draft.

Caner Uzunca performed the experiments, prepared figures and/or tables, and approved the final draft.

İbrahim Bahçivan analyzed the data, prepared figures and/or tables, and approved the final draft.

Ahmed Abdelmoeen Abbass conceived and designed the experiments, prepared figures and/or tables, and approved the final draft.

The following information was supplied relating to ethical approvals (i.e., approving body and any reference numbers):

Research Ethics Committee of Faculty of Medicine Benha University (approval number: 12.7.2022; approval date: 04 September 2022 (MoHPNo:0018122017/CertificateNo:1017)).

The following information was supplied regarding data availability:

The raw data are available in the Supplemental File.

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
