# Peer review of "Relationship between lower extremity strength asymmetry and linear multidimensional running in female tennis players"

_PeerJ, doi:10.7717/peerj.18148_

## Round 0.1 · original submission · Major Revisions

Dear authors.

Reviewer 1 raised some questions regarding the manuscript. Please revise the manuscript considering the suggestions.

Note: The comments of R2 are in their appended PDF

Thank you.

Best regards.

·

Basic reporting

The authors argued that studies examining strength asymmetries and linear and multidimensional running performances of tennis players are insufficient in the literature. However, they need to describe explicitly these few studies and then present the gap that the present study will address.

The English language and several typos need to be addressed.

Experimental design

The present study reports the relationship between horizontal and vertical jump asymmetries with linear and change-of-direction running performances in young tennis players. The authors recruited a good number of participants (n=56), and the study design sounds appropriate. However, the performance tests chosen apparently are not related to tennis performance.

Validity of the findings

The authors argued that studies examining strength asymmetries and linear and multidimensional running performances of tennis players are insufficient in the literature. However, they need to describe explicitly these few studies and then present the gap that the present study will address. Otherwise, examining the potential relationship between jump asymmetry and 10 and 30 sprint performance is irrelevant for tennis players. With the dimensions of a tennis court in mind, I wonder if measuring the 30 m linear sprint performance makes sense.

Additional comments

Introduction
The rationale for recruiting only young tennis players should be addressed.
The rationale for using jump performance as a proxy measure of performance asymmetry should be addressed.
The authors must verify the reference description: numbers vs. authors’ names.

Methods
How was dominance determined?
Why 10 and 30 m sprints are relevant to tennis players?
The author should report body mass (in kg) instead of weight.
Since jumpers usually change body posture between takeoff and landing when performing a vertical jump with arms swing, jump height should not be measured using a jumping mat or any other device applying the flight time method. I suggest the author report the limitation and intrasession reliability to mitigate the potential lack of data validity.
Regarding the assumption of data normality, please clarify if the range of -1.5 to +1.5 was applied for both skewness and kurtosis results.
Pearson correlations are vulnerable to outliers. Please clarify if it was verified. In this regard, I suggest the author present a scatter plot with relationship results.

Results
The author should report that they found a positive relationship between CMJ asymmetry and 30 m *time*; otherwise, the reader may interpret that asymmetry positively correlated with 30 m performance.

Reviewer 2 ·

Basic reporting

No comment

Experimental design

No comment

Validity of the findings

No comment

Additional comments

See comments in the attached PDF

Annotated reviews are not available for download in order to protect the identity of reviewers who chose to remain anonymous.

---

## Round 0.2 · Major Revisions

Dear authors,

Reviewer 1 report indicates the need of some revisions.

Please revise the manuscript considering the suggestions.

Thank you.

Best regards.

·

Basic reporting

Although I recognized the author's effort to address my concerns, IMHO, the gap in the literature the current study would address still needs to be clarified. The authors should make the introduction more objective, not longer.

Experimental design

Please describe explicitly what we know and what we don't know about the relationship between limb asymmetry and performance in tennis players. After that, please describe the originality of the current study.

Validity of the findings

Regarding the performance tests chosen, it would be beneficial to clearly mention studies that have shown a relationship between these field tests and real tennis performance or injury prevalence. If such studies are not available, it could inspire future research to investigate this relationship.

The results might be reliable regarding using the flight time method for CMJ with arms-free testing, but they should not be judged as valid. It must be mentioned.

Please describe that the presence of outliers was verified and refuted.

---

## Round 0.3 · Major Revisions

Dear Authors,

Please consider the suggestions provided by reviewer 1 in this round.

Thank you.

Best regards.

·

Basic reporting

English needs to be improved.

Experimental design

The research question is still problematic.

The text of the introduction is controversial and should not be. The authors mostly describe that "asymmetry increases the risk of injury and impacts performance". However, they also describe Afonso et al. point of view that there is no relationship between asymmetry and injury.

It's important to note that the focus of the study is not on asymmetry and injury but rather on the relationship between asymmetry and sprint and COD performance.

Please also note that the study reports the relationship between jump asymmetry and sprint/COD performance. The author should not describe that the present study "enhances our understanding of how limb asymmetry affects tennis performance" since tennis performance was not measured in this study.

Validity of the findings

The conclusion should be limited to what was found in the study.

---

## Round 0.4 · accepted · Accept

Dear Editor,

We have one suggestion for accept by one of the reviewers. In my opinion, the questions raised by the other author (who indicated that he does not intend to carry out any more reviews) were correctly answered by the authors.

Thank you.

Best regards.